# Results of the CROWN challenge on automated assessment of circle of Willis morphology

**Iris N. Vos**[1]                                             I.N.VOS-6@UMCUTRECHT.NL

**Ynte M. Ruigrok**[2]                                         IJ.M.RUIGROK@UMCUTRECHT.NL

**Hugo J. Kuijf**[1]                                           H.KUIJF@UMCUTRECHT.NL

**on behalf of all organizers and participants of the CROWN challenge 2023**

[1] *Image Sciences Institute, University Medical Center Utrecht, The Netherlands*

[2] *Department of Neurology and Neurosurgery, University Medical Center Utrecht, The Netherlands*

## Abstract

Automated assessment of circle of Willis (CoW) morphology may aid the identification of imaging risk factors associated with intracranial aneurysm (IA) development. However, comparative studies that explore optimal methodological approaches are currently lacking. To systematically compare the performance of automated methods to a clinical reference standard, we initiated a scientific challenge associated with MICCAI 2023. This challenge comprised two tasks: automated classification of CoW anatomical variants and automated prediction of CoW artery diameters and bifurcation angles. The challenge dataset comprised 300 TOF-MRA images for training and 300 for testing, all manually annotated. Method were evaluated using balanced accuracy, mean absolute error, and Pearson correlation coefficient metrics. This short paper will present the first results and evaluation of the challenge. The challenge remains open for future submissions, serving as a benchmark for evaluating methods aimed at assessing CoW morphology.

**Keywords:** circle of Willis, intracranial arteries, biomedical image analysis challenge, classification, quantification, magnetic resonance angiography

## 1. Introduction

Bifurcations of the arteries of the circle of Willis (CoW), an arterial anastomotic circle that provides blood to the brain, are a typical location for intracranial aneurysms (IAs) to develop. Variations in CoW configuration, artery diameters, and bifurcation angles may affect hemodynamic stress, contributing to IA development. IAs affect approximately 3% of the general population and cause aneursymal subarachnoid hemorrhage (ASAH) upon rupture, a severe type of stroke (Nieuwkamp et al., 2009). Currently, recommended practice involves repeated screening of first-degree family members of ASAH patients; however, this screening process is not optimal (Bor et al., 2014). Discovering predictive risk factors for IA development could improve this process by identifying high-risk individuals during initial screening. Few imaging risk factors have been identified, and the supporting evidence is limited (Vos et al., 2024; Kancheva et al., 2022). To address this gap, we initiated the Circle of Willis Intracranial Artery Classification and Quantification (CROWN) challenge, aimed at comparing various methodologies for automated assessment of CoW morphology (https://crown.isi.uu.nl/). Two tasks were included: classification of the CoW anatomical variant (Task 1), and quantification of CoW artery diameters and bifurcation angles (Task 2). The challenge was associated with the 26th International Conference on Medical Image Computing and Computer Assisted Intervention (MICCAI) in Vancouver, Canada.

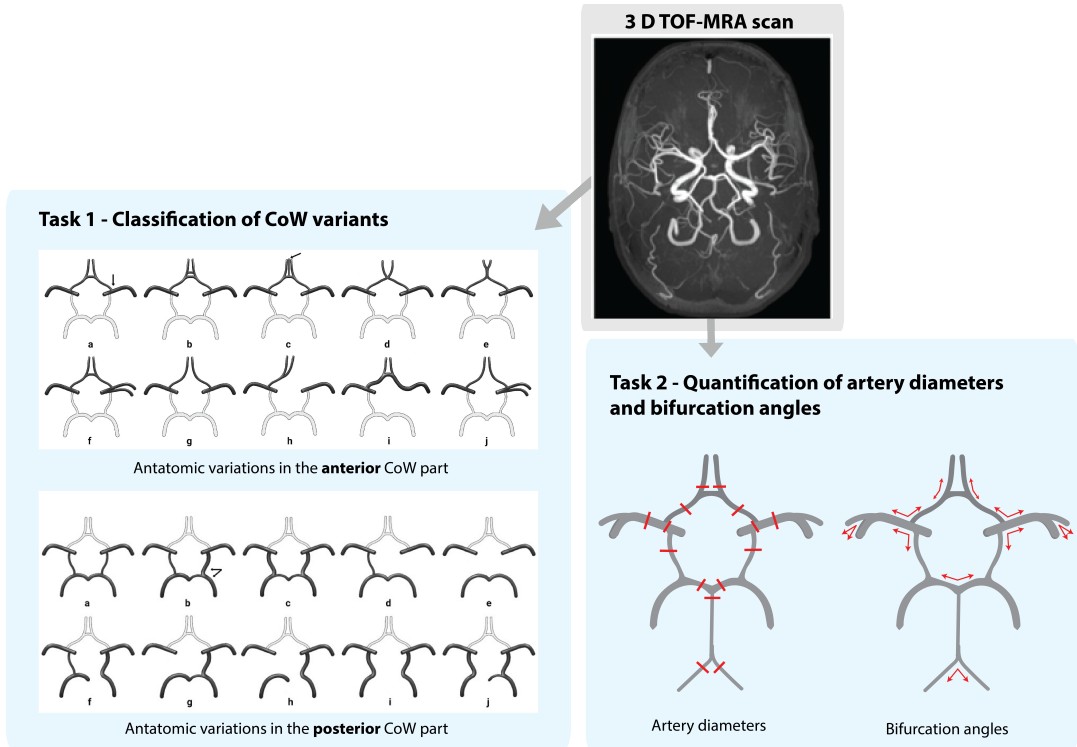

Figure 1: *Upper*: Mean intensity projection of the CoW using 3 D TOF-MRA. *Left*: Classification system for the anterior and posterior CoW part (Lippert and Pabst, 1985), figure adapted from Ophelders et al. (2023). *Right*: Annotations of the artery diameters and bifurcation angles included in the challenge.

## 2. Materials and Methods

Challenge data was obtained from two familial screening cohorts conducted at the University Medical Center Utrecht (Utrecht, the Netherlands). We provided 300 3 D time-of-flight magnetic resonance angiography (TOF-MRA) images from healthy individuals as training data. Another 300 TOF-MRA images were used as test data and kept secret. For Task 1, participants were asked to classify the anterior and posterior CoW part based on an established system (Lippert and Pabst, 1985) that divides each part into 10 classes, see Figure 1. Annotations were performed by two observers. To ensure sufficient training instances, we included only variants with more than 10 occurrences, resulting in five anterior and eight posterior classes. For Task 2, participants were asked to predict the diameter of 15 CoW arteries and the angle of 10 CoW bifurcations. Annotations were performed by three observers using computer-assisted methods.

Evaluation metrics included balanced accuracy (BA) for Task 1; and mean absolute error (MAE) and Pearson correlation coefficient for Task 2. Reported MAE scores were averaged across all predicted artery diameters or bifurcation angles. The evaluation code can be found on github (https://github.com/irisnadinevos/crownchallenge-evaluation-example).

| Rank | Team name | BA (anterior) | BA (posterior) |
|------|-----------|---------------|----------------|
| 1 | **Sibets&USTS** | 0.26 | 0.27 |
| 2* | **AIntropy** | 0.20 | 0.40 |
| 2* | **Labcom I3M** | 0.25 | 0.20 |
| 4 | **agaldran (segs)** | 0.21 | 0.13 |
| 5 | **agaldran (ims)** | 0.20 | 0.14 |
| 6 | **DCS_CUSAT** | 0.20 | 0.11 |

Table 1: Challenge outcome for Task 1 (i.e. automated classifcation of CoW variants) on 300 test cases. BA = balanced accuracy. * denotes shared second place.

| Rank | Team | Artery diameters | | Bifurcation angles | |
|------|------|------------------|---------|--------------------|---------|
| | | MAE (mm) | Pearson | MAE (degrees) | Pearson |
| 1 | **Snaillab** | $0.44 \pm 0.21$ | 0.45 | $28 \pm 5$ | 0.12 |
| 2 | **AIntropy** | $0.50 \pm 0.12$ | 0.36 | $16 \pm 5$ | 0.01 |
| 3 | **Labcom I3M** | $0.87 \pm 0.26$ | 0.14 | $29 \pm 6$ | 0.06 |

Table 2: Challenge outcome for Task 2 (i.e. automated prediction of CoW artery diameters and bifurcation angles) on 300 test cases. MAE = mean absolute error.

Our challenge design (Vos et al., 2023) was submitted to the MICCAI Special Interest Group (SIG) challenges, peer reviewed, and accepted for the MICCAI 2023 series.

## 3. Results and Conclusion

We present the results from teams participating in the CROWN challenge between April and August 2023, during which six teams from five different countries submitted a solution.

For Task 1, the top-3 solutions achieved BA scores between 0.20 and 0.40 for anterior and posterior CoW classification, as summarized in Table 1. The proposed methods employed a graph neural network (team "AIntropy"), atlas-based approach (team "Labcom I3M"), or ResNet50V2 architecture (team "Sibets&USTS").

The proposed solutions to Task 2 yielded average MAE scores between 0.44 and 0.87 mm for artery diameters, whilst the average MAE scores for bifurcation angles were between 16 and 29 degrees. Pearson correlation coefficients ranged from 0.14 to 0.45 for diameter predictions and from 0.01 to 0.12 for angle predictions, see Table 2. All teams relied on voxel-based segmentations to compute diameters or angles, i.e. no end-to-end approaches were proposed.

By organizing the CROWN challenge, we have provided a future benchmark for evaluation of methods to automatically assess CoW morphology. The challenge addresses a complex problem and current outcomes have yet to meet clinical standards. The challenge remains open for future submissions from the MICCAI, MIDL, and related scientific communities, encouraging continued progress in the field.

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
