# OpenReview forum: "Results of the CROWN challenge on automated assessment of circle of Willis morphology"
_MIDL.io/2024/Short_Papers — MIDL 2024 Short Papers_

### Official Review · Reviewer_1KXt · 2024-04-24

**Confidence:** 4
**Final Rating:** 5

**Review:**

The paper describes results from a MICCAI competition on circle of willis (CW) assessment. The CW is very variable across individuals, and classifying variants is a challenging task addressed. Artery diameters and angles are also assessed. Taken together, these could provide quantitative evidence for the risk of aneurysm.


Strengths
- A challenging task, that is very relevant clinically and scientifically
- Very valuable data and annotation work. Congratulations to the challenge organisers
- Interesting observation about no end-to-end approach being used for task 2, surely an avenue for future work

Weaknesses
- for task 1, the number of classes (10) should be specified explicitly, as well as the no information rate (assuming balance, this is 1/10 = 0.1) to put results into context
- Explanation of why 1) typology of CW and 2) angles and diameters are likely contributors to aneurysm risk would be appreciated. What are the plausible hypothesis? is it about turbulent flow? narrowness?

---

### Decision · Program_Chairs · 2024-04-26

Accept